# Conserved nuclear hormone receptors controlling a novel plastic trait target fast-evolving genes expressed in a single cell

Bogdan Sieriebriennikov[ORCID][¤], Shuai Sun, James W. Lightfoot[ORCID], Hanh Witte, Eduardo Moreno, Christian Rödelsperger[ORCID], Ralf J. Sommer[ORCID]*

Department for Integrative Evolutionary Biology, Max Planck Institute for Developmental Biology, Tübingen, Germany

¤ Current address: Department of Biology, New York University, New York, NY, United States of America
* ralf.sommer@tuebingen.mpg.de

**Data Availability Statement:** Raw reads from the RNA-seq experiment have been deposited at the European Nucleotide Archive, study accession

## Abstract

Environment shapes development through a phenomenon called developmental plasticity. Deciphering its genetic basis has potential to shed light on the origin of novel traits and adaptation to environmental change. However, molecular studies are scarce, and little is known about molecular mechanisms associated with plasticity. We investigated the gene regulatory network controlling predatory *vs.* non-predatory dimorphism in the nematode *Pristionchus pacificus* and found that it consists of genes of extremely different age classes. We isolated mutants in the conserved nuclear hormone receptor *nhr-1* with previously unseen phenotypic effects. They disrupt mouth-form determination and result in animals combining features of both wild-type morphs. In contrast, mutants in another conserved nuclear hormone receptor *nhr-40* display altered morph ratios, but no intermediate morphology. Despite divergent modes of control, NHR-1 and NHR-40 share transcriptional targets, which encode extracellular proteins that have no orthologs in *Caenorhabditis elegans* and result from lineage-specific expansions. An array of transcriptional reporters revealed co-expression of all tested targets in the same pharyngeal gland cell. Major morphological changes in this gland cell accompanied the evolution of teeth and predation, linking rapid gene turnover with morphological innovations. Thus, the origin of feeding plasticity involved novelty at the level of genes, cells and behavior.

## Author summary

Rather than following a pre-determined genetic "blueprint", organisms can adjust their development when they perceive relevant environmental signals–a phenomenon called plasticity. This improves performance in changing environment and may also affect how species evolve. To learn how plasticity works on the mechanistic genetic level, we investigated the roundworm *Pristionchus pacificus*. It may develop either as a toothed predator or as a narrow-mouthed microbe-eater depending on food source and population density, an ability that evolved less than 100 million years ago. Previous studies identified switch

number PRJEB34615 (http://www.ebi.ac.uk/ena/data/view/PRJEB34615). Nucleotide sequences of the fragments used to create transgenic constructs are provided in S1 Data. Phylogenetic trees and alignments used to generate them are provided in S2 Data. Data used to generate the boxplot in Fig 2B and the stacked barplot in Fig 3B are provided in S3 and S4 Data, respectively. Landmark coordinates used to conduct geometric morphometric analysis are provided in S5 Data.

**Funding:** This work was supported by institutional funds of the Max-Planck Society (to R.J.S.) and the China Scholarship council (to S.S.). Both funders, the Max-Planck Society and the China Scholarship Council did not play any role in the study design, data collection and analysis, decision to publish, or the preparation of the manuscript.

**Competing interests:** The authors have declared that no competing interests exist.

genes, whose inactivation or overactivation forces either predatory or non-predatory development. Here, we identified the first core gene, which is required for the specification of both morphologies. It encodes a transcription factor, whose inactivation creates animals that appear intermediate between predators and non-predators. We queried which genes are simultaneously controlled by this previously unknown regulator and by a closely related protein that acts as a classical switch. All of the co-regulated genes were recently born and are acting in a single cell that was strongly modified when predator vs. non-predator plasticity evolved. We suggest that conserved regulators of different classes enlisted novel genes in a refurbished cell to regulate a novel plastic trait.

## Introduction

Developmental plasticity is the ability to generate different phenotypes in response to environmental input [1]. As a result, even genetically identical individuals may develop distinct phenotypes, the most extreme example being castes in social insects [2]. Developmental plasticity is attracting considerable attention in the context of adaptation to climate change [3–6] and as a facilitator of evolutionary novelty [7–11]. However, the role of plasticity in evolution has been contentious [6,12] because the genetic and epigenetic underpinnings of plastic traits have long remained elusive. Nonetheless, recent studies have begun to elucidate associated molecular mechanisms in insects and nematodes [13–16]. Ultimately, the identification of gene regulatory networks (GRN) controlling plasticity will provide an understanding of development in novel environments and enable the testing of theories about the long-term evolutionary significance of plasticity.

The free-living nematode *Pristionchus pacificus* has recently been established as a model to study plasticity [13]. These worms can develop two alternative mouth forms, called eurystomatous (Eu) and stenostomatous (St) mouth forms, respectively. Eu morphs have a wide buccal cavity and two large opposed teeth enabling predation on other nematodes, while St morphs have a narrow buccal cavity and one tooth limiting their diet to microbial sources [17,18] (Fig 1A–1C, S1A Fig). The wild-type *P. pacificus* strain PS312 preferentially forms Eu morphs in standard culture conditions on agar plates, but becomes predominantly St in liquid culture [19]. Additionally, nematode-derived modular metabolites excreted by adult animals induce the predatory Eu morph [20,21]. A forward genetic screen identified the sulfatase gene *eud-1* as a developmental switch confirming long-standing predictions that plastic traits are regulated by binary switches [18]. Subsequent studies implicated several other enzyme-encoding genes, such as *nag-1*, *nag-2*, and *sult-1/seud-1* in regulating mouth-form plasticity [22–25]. Additionally, the chromatin modifier genes *lsy-12* and *mbd-2* influence *eud-1* expression [26]. In contrast, only one transcription factor, the nuclear hormone receptor (NHR) NHR-40, was so far found to regulate mouth-form fate [27], and no downstream targets have been identified (Fig 1D).

Here, we leveraged the power of suppressor screen genetics to identify the conserved nuclear hormone receptor NHR-1 as a second transcription factor controlling mouth-form development. It differs from *nhr-40* and all the other genes identified to date in that *nhr-1* mutants develop a morphology that combines features of the two morphs, consistent with disrupted mouth-form determination. Furthermore, transcriptomic profiling revealed that NHR-40 and NHR-1 share transcriptional targets, which exhibit functional redundancy and are expressed in a single pharyngeal gland cell, g1D. This cell has undergone extreme morphological remodeling in nematode evolution, which is associated with the emergence of teeth and

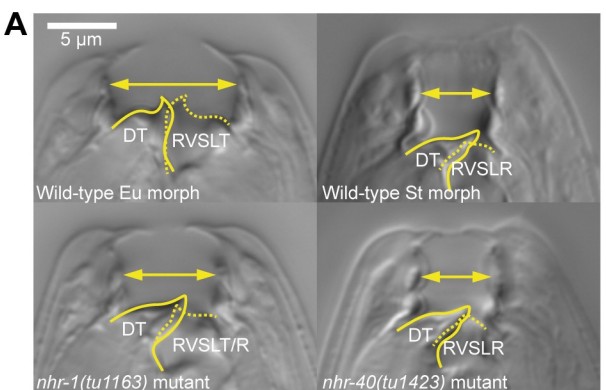
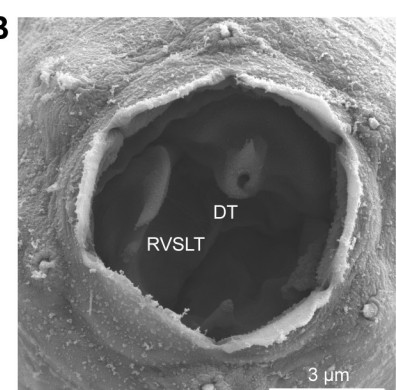
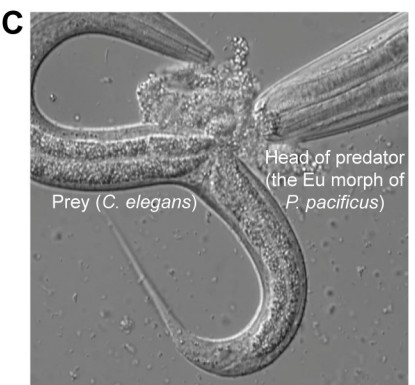
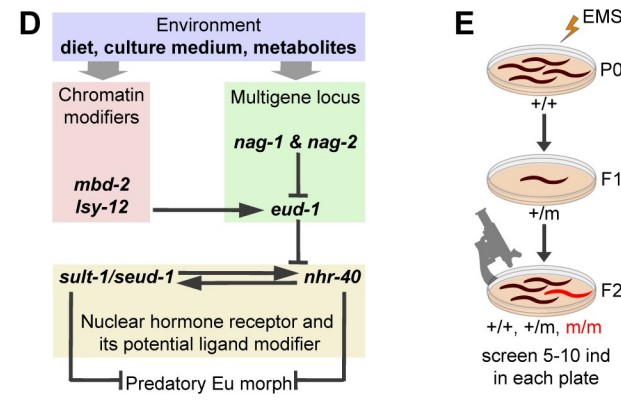

**Fig 1. Mouth-form plasticity in *P. pacificus*.** (A) Mouth structure of wild-type eurystomatous (Eu) morph, wild-type stenostomatous (St) morph, *nhr-1* mutant, and *nhr-40* mutant. Unlabeled images in two focal planes are shown in S1A Fig. (B) Scanning electron microscopy image of the mouth opening of the Eu morph. (C) The Eu morph devouring its prey. (D) Putative gene regulatory network controlling mouth-form plasticity in *P. pacificus*. (E) Design of the suppressor screen. DT = dorsal tooth, RVSLT = right ventrosublateral tooth, RVSLR = right ventrosublateral ridge, EMS = ethyl methanesulfonate.

predatory feeding. Interestingly, *nhr-1* and *nhr-40* are well conserved, whereas all target genes are rapidly evolving and have no orthologs in *C. elegans*. This study enhances the understanding of the GRN regulating mouth-form plasticity, elucidates the evolutionary dynamics of underlying genes and links morphological innovations with rapid gene evolution.

## Results

### Suppressor screen in *nhr-40* identifies another NHR gene regulating mouth-form development

While our previous studies have identified various components involved in the regulation of mouth form plasticity, most of these genes are expressed in neurons responsible for environmental sensing and we had yet to find factors acting in the tissues forming the mouth structure. Therefore, we looked for more downstream factors by conducting a suppressor screen in the mutant background of *nhr-40*. This is the most downstream gene in the current GRN controlling *P. pacificus* mouth-form plasticity and it encodes a transcription factor [27]. We mutagenized *nhr-40(tu505)* worms, which are all-Eu, and isolated one allele, *tu515*, that had a no-Eu phenotype (Fig 1E, Table 1).

**Table 1. Mouth-form frequencies in wild type and mutant lines.**

| Medium | Genotype | Eu, % | N |
|---|---|---|---|
| NGM agar | wild type PS312 | 98 | 650 |
| NGM agar | *nhr-40(tu505)* | 100 | 100 |
| NGM agar | *nhr-40(tu505) tu515* | 0 | 136 |
| NGM agar | *tu515* | 0 | 136 |
| NGM agar | *nhr-1(tu1163)* | 0 | 133 |
| NGM agar | *nhr-1(tu1164)* | 0 | 140 |
| NGM agar | *nhr-1(tu1163)/tu515* | 0 | 70 |
| NGM agar | *nhr-1(tu1163);tuEx305[nhr-1(+);egl-20p::TurboRFP]* | 85 | 110 |
| NGM agar | *nhr-1(tu1163);tuEx310[nhr-1(+);egl-20p::TurboRFP]* | 86 | 112 |
| NGM agar | *nhr-1(tu1163);tuEx328[nhr-1(+)::HA;egl-20p::TurboRFP]* | 86 | 150 |
| NGM agar | *nhr-40(tu505) nhr-1(tu1163)* | 2 | 134 |
| NGM agar | *nhr-40(tu1418)* | 0 | 150 |
| NGM agar | *nhr-40(tu1419)* | 0 | 150 |
| NGM agar | *nhr-40(tu1420)* | 0 | 150 |
| NGM agar | *nhr-40(tu1423)* | 0 | 150 |
| NGM agar | *nhr-40(iub6)* | 100 | 100 |
| NGM agar | *nhr-40(tu1421)* | 100 | 150 |
| NGM agar | *nhr-40(tu1422)* | 100 | 100 |
| NGM agar | duodecuple Astacin mutant[a] | 98 | 55 |
| NGM agar | quintuple CAP mutant[b] | 94 | 50 |
| NGM agar | *PPA04200(tu1213) PPA39293(tu1214)* | 100 | 50 |
| NGM agar | *PPA04200(tu1216) PPA39293(tu1217)* | 100 | 50 |
| NGM agar | *PPA27560(tu1475)* | 100 | 51 |
| NGM agar | *PPA27560(tu1476)* | 100 | 53 |
| NGM agar | *PPA30108(tu1230)* | 100 | 50 |
| NGM agar | *PPA30108(tu1231)* | 100 | 50 |
| NGM agar | *PPA30435(tu1477)* | 100 | 48 |
| NGM agar | *PPA30435(tu1478)* | 98 | 54 |
| NGM agar | *PPA38892(tu1473)* | 100 | 50 |
| NGM agar | *PPA38892(tu1474)* | 100 | 50 |
| S-medium | wild type PS312 | 5 | 850 |
| S-medium | *nhr-40(tu505)* | 100 | 150 |
| S-medium | *nhr-40(tu1418)* | 0 | 150 |
| S-medium | *nhr-40(tu1419)* | 0 | 150 |
| S-medium | *nhr-40(tu1420)* | 0 | 150 |
| S-medium | *nhr-40(tu1423)* | 0 | 150 |
| S-medium | *nhr-40(iub6)* | 100 | 150 |
| S-medium | *nhr-40(tu1421)* | 100 | 150 |
| S-medium | *nhr-40(tu1422)* | 100 | 150 |

N = total number of animals examined

[a]The genotype of the duodecuple Astacin mutant is *PPA03932(tu1259) PPA32730(tu1503);PPA05669(tu1316) PPA05618(tu1317) PPA21987(tu1329) PPA16331(tu1339) PPA27985(tu1340) PPA34430(tu1341) PPA20266(tu1385) PPA42924(tu1386);PPA05955(tu1481) PPA42525(tu1482)*.

[b]The genotype of the quintuple CAP mutant is *tuDf6[PPA21912 PPA29522 PPA21910] tuDf7[PPA05611 PPA39470] tuDf8[PPA13058 PPA39735]*.

The phenotype was fully penetrant, both in the presence of *nhr-40(tu505)* and after out-crossing, *i.e.* Eu animals were never observed under any culture condition. Thus, *tu515* represents a novel factor influencing the mouth-form ratio. Interestingly, however, *tu515* mutants also exhibited a non-canonical mouth morphology (Fig 1A, S1A and S3 Figs). In contrast to all previously isolated mutants, which either display altered mouth-form frequencies or an aberrant morphology, *tu515* individuals develop a morphology that combines normal features of the two morphs with no apparent dimorphism. Specifically, *tu515* mutants closely resemble the St morph in that they have a flattened dorsal tooth, lack a fully developed right ventrosublateral tooth, and the anterior tip of the promesostegostom aligns with the anterior tip of the gymnostom plate. However, the width of the mouth and the curvature of the dorsal tooth appear intermediate between Eu and St, and the right ventrosublateral ridge is frequently enlarged and resembles an underdeveloped tooth of the Eu morph (Fig 1A, S1A Fig). Therefore, while other known mutants affect mouth-form determination by changing the preferred developmental trajectory, *tu515* is the first mutant that disrupts determination, resulting in non-canonical morphology that resembles the St morph but combines features of both morphs.

To map *tu515*, we performed bulked segregant analysis. We examined the list of non-synonymous and nonsense mutations within the candidate region on the X chromosome (S2B Fig, S1 Table) and discovered a non-synonymous mutation in another NHR-encoding gene, *nhr-1*. The substitution changed the sequence of a highly conserved FFRR motif within the DNA recognition helix [28] to FFRW, which may cause the loss of DNA-binding activity. We performed the following experiments to verify that *nhr-1* is the suppressor of *nhr-40(tu505)*. First, we created *nhr-1* mutants using CRISPR/Cas9 by generating frameshift mutations at the beginning of the ligand-binding domain (LBD). The resulting alleles *tu1163* and *tu1164* exhibited a no-Eu phenotype and the same morphological abnormalities as *tu515* (Fig 1A, S1A Fig, Table 1). Second, we crossed the *tu1163* and *tu515* mutants and established that *tu1163/tu515* trans-heterozygotes were no-Eu showing that the two mutants do not complement each other (Table 1). Third, we overexpressed the complementary DNA (cDNA) of *nhr-1* driven by the *nhr-1* promoter region in the *nhr-1(tu1163)* mutant background and obtained an almost complete rescue (Table 1). Fourth, we crossed *nhr-1(tu1163)* with *nhr-40(tu505)* and observed a highly penetrant no-Eu phenotype in double mutant animals, similar to the phenotype of *tu515 nhr-40(tu505)* mutants (Table 1). Taken together, frameshift alleles of *nhr-1* and the original suppressor allele *tu515* exhibit the same phenotype, do not complement each other, and have identical epistatic interactions with *nhr-40(tu505)*. Therefore, we conclude that *nhr-1* is the suppressor of *nhr-40(tu505)*.

## Reverse genetic analysis of *nhr-40* results in all-stenostomatous mutants

The available alleles of *nhr-1* and *nhr-40* have different phenotypes with regard to mouth-form frequency and morphology. This is surprising because NHRs often form heterodimers [29], in which case *loss-of-function* phenotypes of interacting partners are identical. Two different hypotheses could explain our observations. First, *nhr-1* and *nhr-40* may indeed have different functions. Second, the three available alleles of *nhr-40* (*tu505*, *iub6*, *iub5*), all of which are non-synonymous substitutions outside of the DNA-binding domain (DBD) [27], may represent *gain-of-function* alleles. Our previous analysis had suggested that these alleles are *loss-of-function* based on the phenotype of *nhr-40* overexpression, which resulted in all-St animals [27]. However, we recently realized that in *C. elegans*, overexpression of *Cel-nhr-40* and *loss-of-function* of *Cel-nhr-40* induced by RNAi and a deletion mutation all cause similar developmental defects [30]. This may occur if NHR-40 inhibits its own transcription [31] or if the

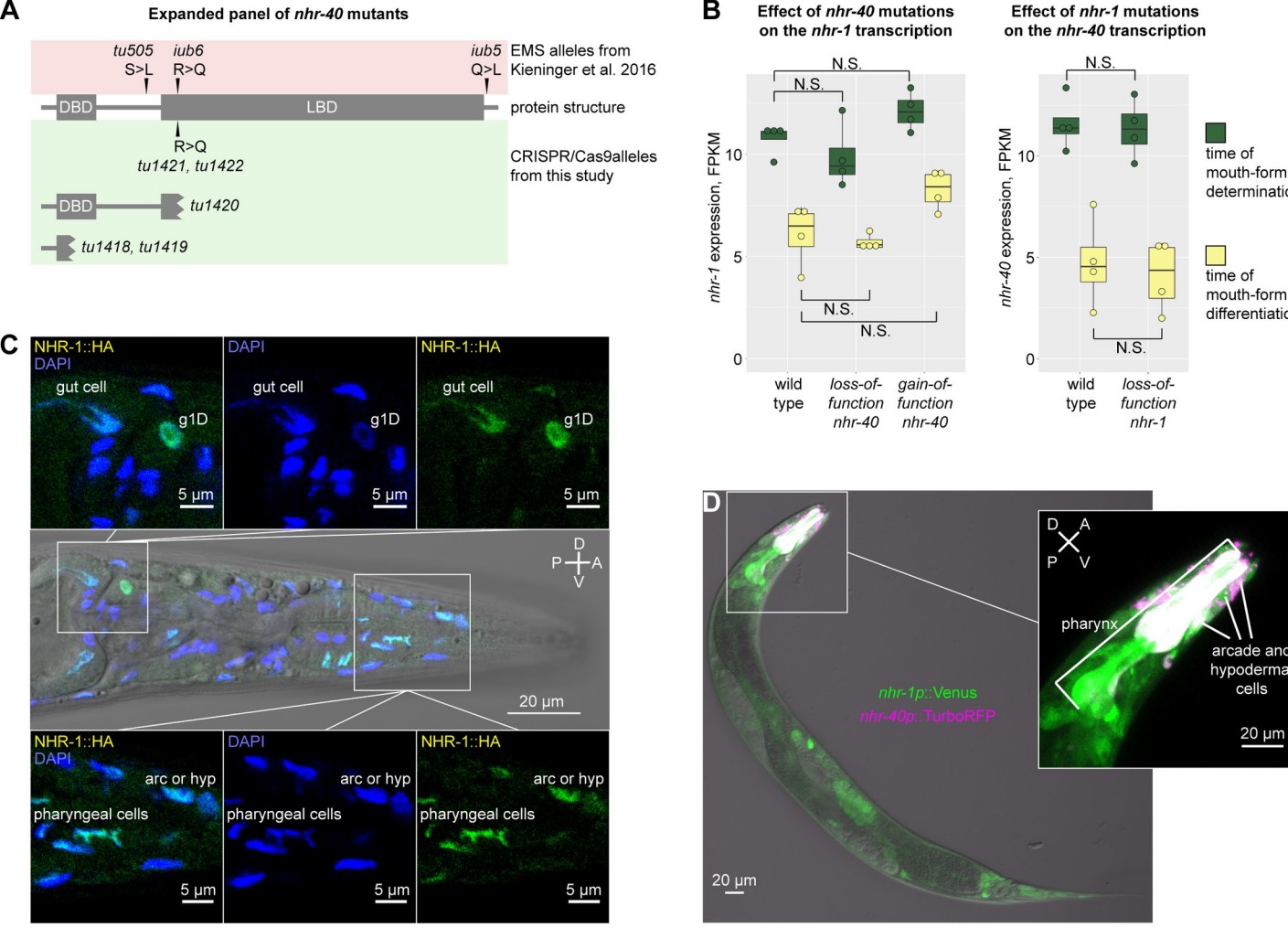

**Fig 2. Reverse genetics, transcriptomics and expression patterns of *nhr-40* and *nhr-1*.** (A) Protein structure of NHR-40 in wild-type and mutant animals. (B) Expression levels of *nhr-40* and *nhr-1* in wild type and mutants as revealed by transcriptomic profiling. (C) Antibody staining against the HA epitope in an *nhr-1* rescue line. (D) Expression patterns of *nhr-40* and *nhr-1* transcriptional reporters in a double reporter line. TurboRFP (magenta) and Venus (green) channels are presented as maximum intensity projections. Co-expression results in white color. D = dorsal, V = ventral, A = anterior, P = posterior, N.S. = not significant, FPKM = Fragments Per Kilobase of transcript per Million mapped reads.

concatenated coding sequence of the rescue construct acts as a substrate to induce RNAi [30]. Therefore, we investigated *nhr-40* in *P. pacificus* further, and generated nonsense alleles using CRISPR/Cas9.

We introduced mutations in two different locations in *nhr-40* (Fig 2A). The alleles *tu1418* and *tu1419* truncate the DBD. The *tu1420* allele contains a frameshift at the beginning of the LBD while leaving the DBD intact. We phenotyped the newly obtained mutants in liquid S-medium, which represses the Eu morph, and on agar plates, which induces it [19]. All frame-shift alleles had a completely penetrant all-St phenotype in both culture conditions, which is opposite to the original ethyl methanesulfonate (EMS) alleles (Table 1). The newly obtained *nhr-40* mutants displayed no morphological abnormalities such as those observed in *nhr-1* mutants. Moreover, *nhr-1* mutants were epistatic to such mutants with respect to abnormal mouth morphology (S4 Fig). Additionally, we created a *null* allele, *tu1423*, which contains a 13 kb deletion or rearrangement of the locus (S2A Fig). This *null* allele again had a completely

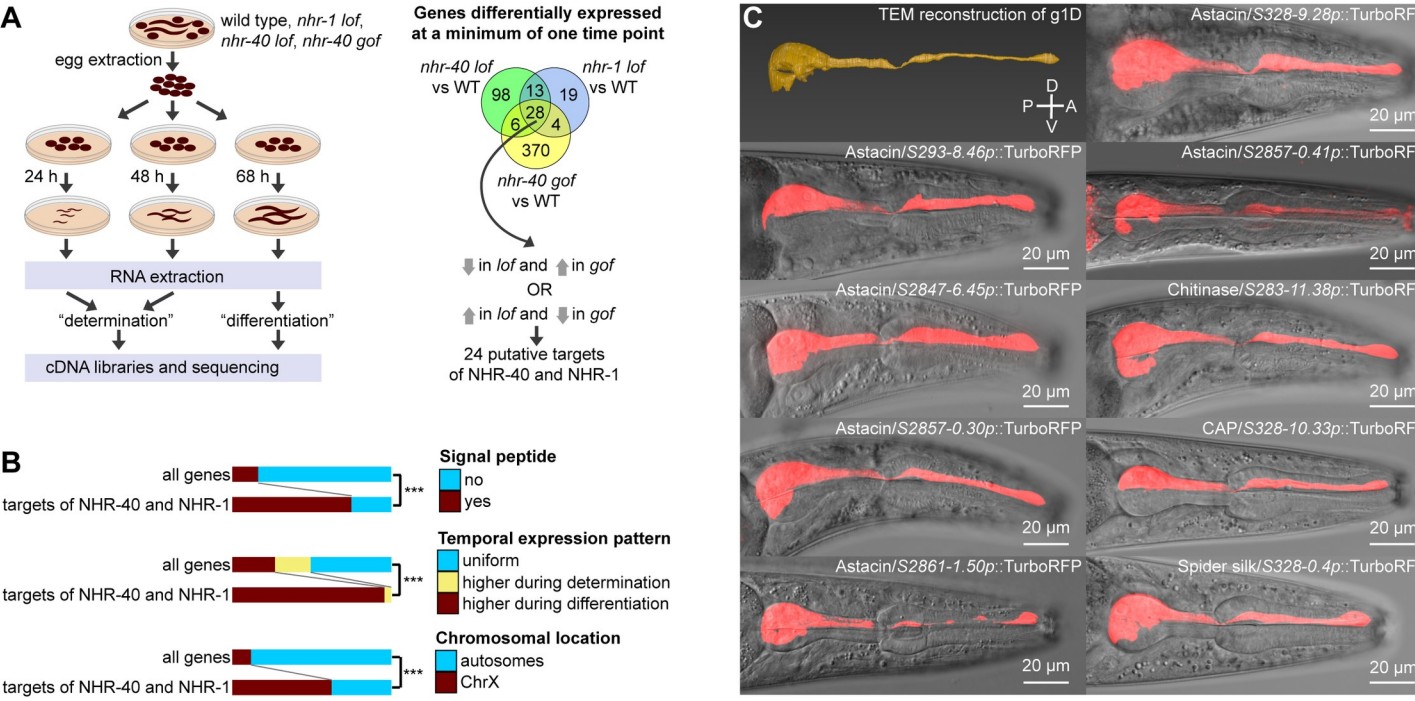

**Fig 3. Target genes of NHR-40 and NHR-1.** (A) Experimental setup of transcriptomics experiment and selection criteria to identify target genes. (B) Trends among target genes compared to genome-wide pattern. (C) Transmission electron microscopy reconstruction of the dorsal pharyngeal gland cell (g1D) [53] and expression patterns of transcriptional reporters for nine selected targets of NHR-40 and NHR-1. TurboRFP channel is presented as standard deviation projections. *lof = loss-of-function*, *gof = gain-of-function*, *** = p<0.001, D = dorsal, V = ventral, A = anterior, P = posterior.

penetrant all-St phenotype and showed no morphological abnormalities (Table 1). To eliminate the possibility that the phenotype of the EMS mutants was caused by random mutations outside *nhr-40*, we introduced a nucleotide substitution identical to *iub6* via homology-directed repair (Fig 2A). Indeed, the two resulting alleles, *tu1421* and *tu1422*, had an all-Eu phenotype, identical to that of *iub6* and other EMS alleles, and opposite to that of the frameshift alleles (Table 1). Thus, frameshift mutations in DBD, LBD, and the deletion/rearrangement of the entire gene have an opposite phenotype to that of the three previously isolated non-synonymous substitutions. We conclude that *tu505*, *iub6*, *iub5*, *tu1421* and *tu1422* are *gain-of-function* alleles.

## NHR-40 and NHR-1 interact post-transcriptionally

In GRNs, transcription factors may activate or repress each other transcriptionally [32–36], or alternatively, they may interact at the post-transcriptional level. The latter includes indirect interactions, such as independent binding to the same promoters [37], or ligand-mediated interactions [38]. To distinguish if *nhr-1* and *nhr-40* interact at the transcriptional or post-transcriptional level, we analyzed the transcriptomes of wild type, *nhr-1 loss-of-function*, *nhr-40 loss-of-function* and *nhr-40 gain-of-function* mutants at two developmental stages (Fig 3A). RNA collected from J2-J4 larvae is enriched with transcripts expressed at the time of mouth-form determination, as environmental manipulation during this time window affects morph frequency [39]. RNA collected from J4 larvae and adults is enriched with transcripts expressed at the time of mouth-form differentiation, because cuticularized mouthparts that distinguish the two morphs are believed to be secreted during the J4-adult molt [40]. We found that at both time points, *nhr-40* transcript levels were not affected by *loss-of-function* of *nhr-1*.

Similarly, *nhr-1* transcript levels were not affected by *loss-of-function* of *nhr-40*, although they were slightly, but not significantly increased by *nhr-40 gain-of-function* (Fig 2B). Thus, at the transcriptional level, both *nhr* genes remain unaffected by the *loss-of-function* of the other gene. Therefore, NHR-40 and NHR-1 may interact at the post-transcriptional level, although the possibility remains that their transcriptional interaction in specific cells is masked in whole-animal transcriptome data.

### *nhr-40* and *nhr-1* are expressed at the site of polyphenism

Next, we wanted to determine the expression pattern of *nhr-1* and *nhr-40* and test if they were co-expressed. We took three complementary approaches to establish the expression pattern of *nhr-1*. First, we created transcriptional reporters comprising the presumptive promoter region upstream of the potential start site in the second exon fused with TurboRFP or Venus. The resulting expression pattern was broad with the strongest expression in the head, including both muscle and gland cells of the pharynx, and what may be the hypodermal and arcade cells (Fig 2D, S1B Fig). Second, we performed antibody staining against an HA epitope tag in the *nhr-1* rescue line described above. We observed a similar expression pattern that was predictably localized to the nuclei (Fig 2C). Finally, we used CRISPR/Cas9 to "knock in" an HA tag in the endogenous *nhr-1* locus at the C-terminus of the coding sequence. Antibody staining against HA revealed a similar expression pattern, but with a weaker signal due to the lower number of copies of endogenous DNA (S1C Fig). Together, these results show that NHR-1 localizes to nuclei of multiple cells in the head region, with strong expression in pharyngeal muscle cells, which presumably secrete structural components of the teeth.

To explore whether NHR-40 and NHR-1 are expressed in overlapping tissues, we created a double reporter line, in which the *nhr-40* promoter is fused to TurboRFP and the *nhr-1* promoter to Venus. We observed a strong and consistent expression of *nhr-40* in the head. Specifically, it localized to the pharyngeal muscle cells and cells whose cell body position is consistent with them being arcade or hypodermal cells (Fig 2D, S1D Fig). *nhr-40* and *nhr-1* signals co-localized in a subset of presumptive hypodermal and arcade cells, and in the pharyngeal muscles. In contrast, only *nhr-1* was expressed in the dorsal pharyngeal gland cell g1D (Fig 2D, S1D and S1E Fig). In summary, while the expression of *nhr-40* is more restricted than the expression of *nhr-1*, the two genes display robust co-localization in several cell types.

### Common transcriptional targets of NHR-40 and NHR-1 encode extracellular proteins expressed during mouth-form differentiation

Since NHR-40 and NHR-1 are co-expressed and regulate the same phenotype, we speculate that they regulate a set of common target genes, even though such regulation may be indirect. We analyzed the full list of genes differentially expressed between the wild type and mutant samples from the experiments described above. Given the pleiotropic action of NHR-40 and NHR-1, we applied the following selection criteria. We only retained genes whose transcript levels at either of the two examined time points were simultaneously altered in *nhr-1*, *nhr-40 loss-of-function*, and *nhr-40 gain-of-function* mutants (Fig 3A). Only 28 genes satisfied this criterion, and their expression changed in the same direction in the *loss-of-function* mutants of *nhr-1* and *nhr-40*. We further retained those genes whose expression changed in one direction in the *loss-of-function* mutants of *nhr-1* and *nhr-40*, and in the opposite direction in the *gain-of-function* mutants of *nhr-40* (Fig 3A), resulting in a list of 24 genes, provided in Table 2, Interestingly, the expression of 23 of them decreased in the *loss-of-function* mutants (Table 2).

**Table 2. List of targets of NHR-40 and NHR-1.**

| Chr | Wormbase WS268 identifier | El Paco annotation v1 identifier | Predicted PFAM domains | LFC, *nhr-1* vs. WT, DT | LFC, *nhr-1* vs. WT, DF | LFC, *nhr-40 lof* vs. WT, DT | LFC, *nhr-40 lof* vs. WT, DF | LFC, *nhr-40 gof* vs. WT, DT | LFC, *nhr-40 gof* vs. WT, DF |
|---|---|---|---|---|---|---|---|---|---|
| X | PPA05669 | UMM-S328-9.28-mRNA-1 | Astacin | -6.2 | -8.0 | -7.2 | -7.8 | 2.1 | 1.1 |
| IV | PPA42525 | UMM-S2847-7.46-mRNA-1 | Astacin | -4.8 | -4.0 | -7.8 | -4.7 | 1.4 | NS |
| IV | PPA05955 | UMM-S2847-6.45-mRNA-1 | Astacin | -4.0 | -3.7 | -6.6 | -6.0 | 1.6 | 1.1 |
| X | PPA05618 | UMM-S328-7.47-mRNA-1 | Astacin | -3.6 | -2.9 | -7.7 | -7.0 | 1.5 | 1.0 |
| X | PPA16331 | UMA-S293-8.46-mRNA-1 | Astacin | -2.7 | -4.1 | -3.5 | -4.2 | 1.7 | NS |
| X | PPA39735 | UMM-S328-10.33-mRNA-1 | CAP | -2.2 | -2.5 | -5.1 | -5.3 | 1.4 | NS |
| I | PPA32730 | UMM-S57-4.91-mRNA-1 | Astacin | -2.2 | -1.6 | -3.7 | -3.0 | 1.7 | 0.8 |
| X | PPA13058 | UMM-S328-10.78-mRNA-1 | CAP | -2.1 | -2.4 | -4.8 | -4.8 | 1.4 | NS |
| IV | PPA39293 | UMM-S283-11.38-mRNA-1 | Glyco_hydro_18 | -1.8 | -1.0 | -3.0 | -1.9 | 0.9 | NS |
| X | PPA29522 | UMM-S322-3.5-mRNA-1 | CAP | -1.6 | NS | -3.1 | -1.5 | 1.0 | NS |
| X | PPA39470 | UMM-S293-11.30-mRNA-1 | CAP | -1.5 | -2.2 | -4.4 | -4.9 | 1.4 | NS |
| X | PPA21910 | UMA-S322-3.38-mRNA-1 | CAP | -1.4 | NS | -2.3 | -1.2 | 0.9 | NS |
| IV | PPA04200 | UMM-S283-11.45-mRNA-1 | Glyco_hydro_18; MFS_1 | -1.3 | -0.8 | -2.1 | -1.3 | 0.9 | NS |
| X | PPA21987 | UMA-S322-7.39-mRNA-1 | Astacin | -1.1 | -1.1 | -1.5 | NS | 0.9 | 0.9 |
| X | PPA27985 | UMS-S2861-1.50-mRNA-1 | Astacin | -1.0 | -1.6 | -1.4 | -1.8 | 0.9 | 0.9 |
| X | PPA30108 | UMS-S328-0.4-mRNA-1 | none | -0.9 | -2.0 | -1.7 | -3.1 | 1.4 | 0.7 |
| II | PPA27560 | UMS-S10-46.25-mRNA-1 | none | NS | -2.5 | NS | -1.5 | 1.6 | NS |
| I | PPA30435 | UMM-S57-36.5-mRNA-1 | Lectin_C | NS | -2.5 | -6.4 | -7.5 | 1.7 | 0.8 |
| X | PPA34430 | UMA-S2861-1.27-mRNA-1 | Astacin | NS | -2.0 | -1.4 | -2.2 | 1.0 | 0.9 |
| X | PPA38892 | UMM-S250-3.76-mRNA-1 | ShK | NS | -2.0 | -1.8 | -2.2 | 1.0 | NS |
| X | PPA20266 | UMM-S2857-0.30-mRNA-1 | Astacin | NS | -1.9 | -1.4 | -2.5 | 1.1 | NS |
| X | PPA42924 | UMM-S2857-0.41-mRNA-1 | Astacin | NS | -1.1 | NS | -1.7 | 0.8 | NS |
| I | PPA03932 | UMM-S7-5.16-mRNA-1 | Astacin | NS | -1.1 | NS | -1.6 | 1.1 | 0.8 |
| IV | PPA06264 | UMA-S2838-46.74-mRNA-1 | adh_short; KR; THF_DHG_CYH_C | NS | 2.2 | NS | 3.0 | -2.0 | NS |

Chr = chromosome, LFC = log fold change, WT = wild type, *lof = loss of function*, *gof = gain of function*, DT = determination, DF = differentiation, NS = not significant.

We hypothesized that if the making of cuticularized mouthparts involves these genes, they must encode extracellular proteins, and their expression is likely to be biased towards the time of mouth-form differentiation. To verify the extracellular function of the target proteins, we predicted signal peptides and compared the list of targets with the genome-wide pattern. Indeed, we found that the targets of NHR-40 and NHR-1 are significantly enriched with genes containing signal peptides (Fig 3B). To examine a potential temporal expression bias, we compared the wild-type transcriptomes at the time of mouth-form determination and mouth-form differentiation. While most genes in the genome (51%) showed uniform expression at the two time points, 23 of the 24 targets of NHR-40 and NHR-1 were more highly expressed at the time of mouth-form differentiation (Fig 3B). Surprisingly, we also observed a third trend in our data set. While only 12% of all genes in the genome are located on the X chromosome, 15 of the 24 targets of NHR-40 and NHR-1 were X-linked (Fig 3B), which resembled the bias in the chromosomal location of hermaphrodite-specific somatically expressed genes in *C. elegans* [41].

To explore the potential functions of the NHR-40 and NHR-1 targets, we used information about their annotated protein domains. Surprisingly, 12 of the 24 genes contain an Astacin domain (Table 2) typical of secreted or membrane-anchored Zinc-dependent endopeptidases [42]. Of the 40 Astacin-containing genes in *C. elegans*, only *dpy-31*, *nas-6* and *nas-7* have known functions, whereby mutations in these genes result in abnormal cuticle synthesis [43,44]. Another five of the 24 NHR targets encode a CAP (**c**ysteine-rich secretory proteins, **a**ntigen 5, and **p**athogenesis-related 1) domain (Table 2), which is contained in extracellular proteins with diverse functions [45–47], including the proteolytic modification of extracellular matrix [48]. Two genes belong to the glycoside hydrolases family 18 (Table 2), which includes chitinases and chitinase-like proteins [49] that may modify the cuticle, as chitin is the main component of the cuticle in nematodes [50]. Finally, the NHR target list includes an unannotated protein, PPA30108 (Table 2), which contains multiple GGF and GGR repeats, similar to some structural proteins of spider silk [51,52]. Thus, the examination of the domain composition of the targets of NHR-40 and NHR-1 suggests that many encode enzymes that may directly modify the cuticle.

## A duodecuple Astacin mutant shows no mouth-form abnormalities

Next, we tested if mutations in the identified genes affected mouth-form frequency or morphology. We therefore performed systematic CRISPR/Cas9 knockout experiments of the 23 genes downregulated in the *loss-of-function* mutants. To compensate for potential redundancy between paralogous genes encoding identical domains, we produced lines in which all such genes are inactivated simultaneously. For example, rather than generating 12 strains with mutations affecting single Astacin-encoding genes, we produced a duodecuple mutant line, in which we sequentially knocked out all 12 genes (Table 1). We phenotyped the mutants both on agar plates and in liquid S-medium. However, we detected no significant change in mouth-form frequencies and no recapitulation of the morphological defects of *nhr-1*. Similarly, we produced a quintuple CAP mutant and double chitinase mutants and observed no change in mouth-form frequency or morphology (Table 1). We speculate that this may be caused by the extreme redundancy in the factors involved. For instance, despite mutagenizing 12 Astacin-encoding genes, there are more than 60 such genes in the genome. Consistent with this, in a phenotypic screen of Astacin genes in *C. elegans*, the majority showed no detectable phenotypes and the function of one, *nas-7*, was only elucidated due to its enhancement of a weakly penetrant allele of *nas-6* [44]. Alternatively, it is also possible that some examined genes function in other tissues unrelated to mouth morphology. Therefore, we next studied the spatial expression of selected downstream target genes.

## Downstream targets genes are expressed in the same pharyngeal gland cell

We selected six of the 12 Astacin genes, one chitinase gene, one CAP gene, and the gene bearing similarity to spider silk proteins, and created transcriptional reporters by fusing their promoters with TurboRFP. Remarkably, all reporter lines showed expression in the same single cell, the dorsal pharyngeal gland cell g1D (Fig 3C). In contrast, we found no expression in the pharyngeal muscles or other expression foci of *nhr-40* and *nhr-1*. Thus, all analyzed targets are co-expressed with *nhr-1* in g1D (Fig 3C, S1E Fig). The recent reconstruction of the pharyngeal gland cell system of *P. pacificus* [53] revealed that the cell body of g1D is located at the posterior end of the pharynx. It sends a long process through the entire pharynx to the anterior tip where it connects, via a short duct in the cuticle, to a channel in the dorsal tooth which opens into the buccal cavity (Figs 1B and 3C). Importantly, the process of g1D is surrounded by pharyngeal muscle cells which directly underlie the teeth. Therefore, we hypothesize that the enzymes excreted from g1D act on the structural components that are themselves secreted by the pharyngeal muscles.

## Expansion of the pharyngeal gland cells is concomitant with the emergence of teeth

The expression of the targets of NHR-40 and NHR-1 in g1D is remarkable, because g1D is the site of a major evolutionary innovation in the family Diplogastridae, to which *P. pacificus* belongs. The pharynx in free-living nematodes of the order Rhabditida and the outgroup [54] family Teratocephalidae is divided into two parts. The anterior part, called the corpus, is muscular, and in some lineages ends with a dilation, called the median bulb. The posterior part, called the postcorpus, is divided into a narrow isthmus and a dilation, called the terminal bulb, which contains muscle cells and three to five gland cells. The terminal bulb contains muscular valves that form a specialized cuticular structure, the grinder, which helps fragment food particles [55] (Fig 4). Phylogenetic reconstruction indicates that the outgroup Teratocephalidae, and the rhabditid families Cephalobidae and Rhabditidae retained the ancestral character states, whereby they have a grinder, but no teeth [56–58]. In contrast, Diplogastridae have no grinder, but they have concomitantly gained teeth at the base of the family [7,59]. The acquisition of teeth and the loss of the grinder were accompanied by the reduction of the muscle cells

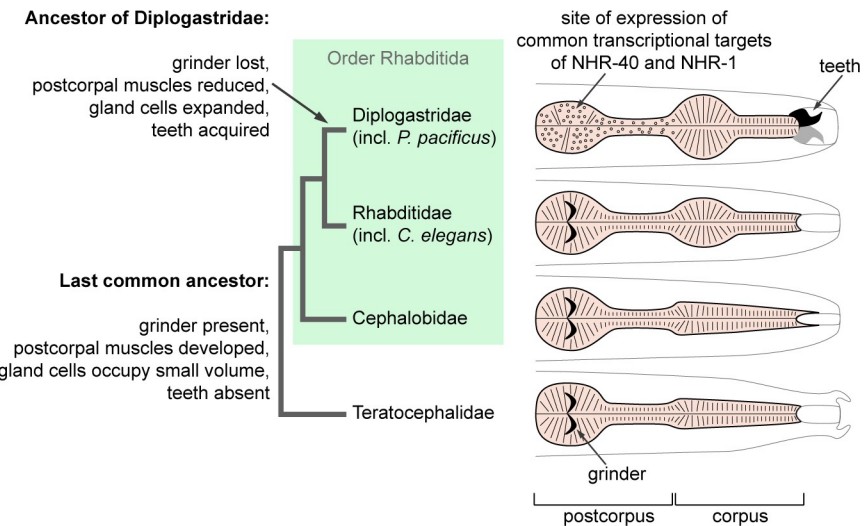

**Fig 4. Evolution of pharynx morphology in the order Rhabditida.**

in the postcorpus, and an expansion of three gland cells g1D, g1VL, and g1VR, one in each sector of the trilaterally symmetrical pharynx [53,59] (Fig 4). While the exact role of pharyngeal gland cells in *C. elegans* and other nematodes has remained elusive [55], we speculate that the functional remodeling of g1D, in which the target genes of NHR-40 and NHR-1 are expressed, may be a prerequisite for the formation of teeth and the evolution of predation. Therefore, we investigated the evolutionary dynamics of the identified genes expressed in this cell.

### Conserved transcription factors regulate fast-evolving target genes

To investigate if the morphological lineage-specific evolutionary innovation in *P. pacificus* and Diplogastridae is associated with taxonomically restricted genes, we reconstructed the phylogeny of NHR genes and their identified targets. This is an important evolutionary question as recent genomic studies involving deep taxon sampling revealed high evolutionary dynamics of novel gene families in *Pristionchus*, with only one third of all genes having 1:1 orthologs between *P. pacificus* and *C. elegans* [60,61]. First, we reconstructed the phylogeny of NHR genes. We identified similar numbers of NHR genes in the genomes of *P. pacificus* and *C. elegans*—254 and 266 genes, respectively. In the phylogenetic tree (Fig 5A), most clades contained genes from predominantly or exclusively one of the two species. These genes likely result from lineage-specific duplications and losses, a phenomenon commonly seen in nematode gene families [62]. *nhr-40* and *nhr-1*, however, belonged to one of the few clades that contained a mixture of genes from both species, with many genes displaying a 1:1 orthology relationship. Indeed, the *P. pacificus* and *C. elegans* copies of *nhr-40* and *nhr-1* showed 1:1 orthology with 100% bootstrap support (Fig 5A). Importantly, *nhr-40* and *nhr-1* are also extremely closely related to each other (Fig 5A). Thus, in the overall context of NHR evolution, *nhr-40* and *nhr-1* are closely related duplicates that have been conserved since the divergence of *P. pacificus* and *C. elegans*.

The conservation of *nhr-40* and *nhr-1* is in stark contrast to the evolutionary history of their downstream targets. To reconstruct the phylogenies of the Astacin, CAP and chitinase genes (Fig 5B–5D), we used functional domains rather than complete genes to facilitate the alignment of genes with different domain architectures. Similar to the case of NHRs, all three gene families exhibit strong signatures of lineage-specific expansions. Furthermore, all target genes containing Astacin, CAP and chitinase domains belonged to such lineage-specific clades (Fig 5B–5D). These findings suggest that the targets of NHR-40 and NHR-1 undergo rapid turnover. This is further supported by the phylogeny of CAP genes within the genus *Pristionchus*. Specifically, the five targets identified in *P. pacificus* clustered separately from the homologs in the early branching species *P. fissidentatus* with 94% bootstrap support (Fig 5E). Thus, two conserved NHRs target rapidly evolving downstream genes of multiple gene families. We speculate that the striking co-expression of the target genes might results from an ancient regulatory linkage between the NHRs and the promoters of the ancestral target genes. Alternatively, however, these transcription factors might have captured new promoters by mutations creating binding sites for them, a possibility that will become directly testable once the target sequences of NHR-1 and NHR-40 have been identified.

### Discussion

In this study, we expanded the GRN controlling predatory *vs.* non-predatory plasticity in *P. pacificus*, thereby enhancing the molecular understanding of plasticity. We uncovered novel genetic factors and genomic features at two regulatory levels, which allowed linking rapid gene evolution with morphological innovations associated with plasticity. First, we identified a mutation in the nuclear receptor gene *nhr-1*, which disrupts mouth-form determination. Most

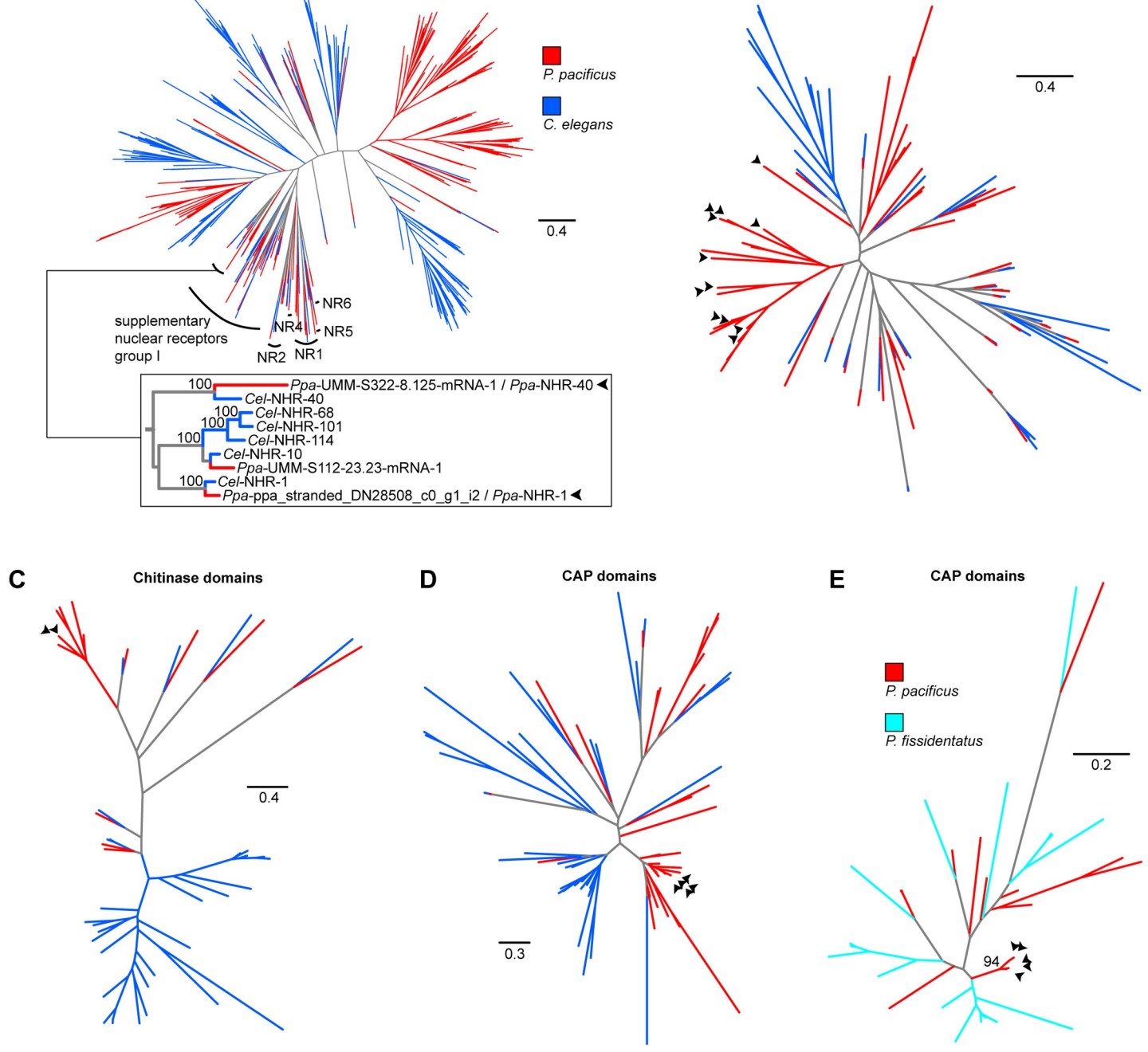

**Fig 5. Evolution of *nhr-40*, *nhr-1*, and their target genes. Arrowheads point at the genes of interest.** Protein-based trees of NHR genes (**A**), Astacin domains (**B**), chitinase domains (**C**), and CAP domains (**D**) in *P. pacificus* and *C. elegans*. (**E**) Nucleotide-based tree of the CAP domains from a poorly-resolved protein-based subtree of all predicted CAP domains in *P. pacificus* and *P. fissidentatus*.

previously identified genes, such as *eud-1* or *sult-1/seud-1*, influence the determination process by affecting the preferred developmental trajectory, but the resulting morphology exhibits no observable differences to the corresponding wild-type morphology [18,23,24]. On the other hand, interfering with heat shock protein activity, including a mutation in *daf-21*/Hsp90, produces aberrant morphologies while maintaining the dimorphism [63]. In contrast to both

classes of genetic interventions, mutations in *nhr-1* lead to a morphology that combines features of normal Eu and St morphs, with no apparent dimorphism (S3 Fig). Therefore, we speculate that NHR-1 is required for mouth-form determination and the specification of both morphs. By contrast, we showed that *gain-* and *loss-of-function* mutations in *nhr-40* result in all-Eu and all-St phenotypes respectively, reminiscent of the role of *daf-12*, another *nhr* gene, in controlling dauer plasticity in *C. elegans* [64]. Different phenotypic effects of *nhr-1* and *nhr-40* are also consistent with the lack of evidence of transcriptional regulation of one factor by the other. Except for DAF-12 in *C. elegans*, no single nematode NHR has been de-orphanised. Therefore, the identification of the potential ligands of NHR-1 and NHR-40 may reveal additional layers of regulation and elucidate their cross-talk. Indeed, recent studies suggested that cytosolic sulfotransferases, including *sult-1/seud-1* in *P. pacificus* and its homolog *ssu-1* in *C. elegans*, may regulate NHRs by modifying their ligands [23,24,65].

Second, the transcriptomic analysis of *nhr-1* and *nhr-40* mutants revealed an unexpectedly small number of downstream targets. While cell-specific signals may be masked in whole-animal transcriptome data, and our selection criteria excluded genes affected by the *gain-of-function* of *nhr-40* in other ways than by exhibiting increased transcript levels, having a small list of target genes enabled a systematic analysis of their genomic location, function and expression. Surprisingly, the majority of targets were located on the X-chromosome, which parallels X-linkage of many previously identified genes associated with mouth-form, including both *nhr-40* and *nhr-1*, and additionally the multigene switch locus comprising *eud-1*, *nag-1* and *nag-2*. While the exact meaning of this phenomenon remains unclear, the X chromosome in *C. elegans* is enriched with hermaphrodite-biased somatically expressed genes [41]. Accordingly, the incidence of Eu morphs is higher in *P. pacificus* hermaphrodites than in males [39], which may be reflected in the chromosomal distributions of the genes associated with the Eu morph.

Both the absence of phenotypes in duodecuple and quintuple mutants, and the restricted expression of all tested genes in the same cell g1D are compatible with extreme functional redundancy. Such redundancy might result from features of genome evolution that are common to nematodes and other animals. Studies over the last decade revealed that nematode genomes are gene-rich and exhibit high rates of gene birth and death [60,66,67]. In particular, enzyme-encoding genes are subject to high evolutionary dynamics [62]. Therefore, the position of genes in GRNs may determine the speed and direction of their evolution. Consistent with this idea, many genes encoding proteins of signal transduction and their terminal transcription factors are highly conserved across animals [68–70]. In this study, we complement this knowledge by showing that the downstream targets of conserved transcription factors are indeed fast evolving genes. Importantly, their expression focus, the g1D cell, also underwent a major evolutionary change, whereby its structural and functional remodeling accompanied the emergence of teeth in the family Diplogastridae. Thus, our study demonstrates that fast-evolving genes are expressed in a fast-evolving cell, linking morphological innovations with rapid gene evolution.

## Materials and methods

### Maintenance of worm cultures and genetic crosses

Stock cultures of all strains used in this study were reared at room temperature (20–25°C) on nematode growth medium (NGM) (1.7% agar, 2.5 g/L tryptone, 3 g/L NaCl, 1 mM $CaCl_2$, 1 mM $MgSO_4$, 5 mg/L cholesterol, 25 mM $KPO_4$ buffer at pH 6.0) in 6 cm Petri dishes, as outlined in the *C. elegans* maintenance protocol [71]. *Escherichia coli* OP50 was used as food source. Bacteria were grown overnight at 37°C in L Broth (10 g/L tryptone, 5 g/L yeast extract, 5 g/L NaCl, pH adjusted to 7.0), and 400 μL of the overnight culture was pipetted on NGM

agar plates and left for several days at room temperature to grow bacterial lawns. *P. pacificus* were passed on these lawns and propagated by passing various numbers of mixed developmental stages. To cross worms, agar plates were spotted with 10 μL of the *E. coli* culture, and five to six males and one or two hermaphrodites were transferred to the plate and allowed to mate. Males were removed after two days of mating.

## Mouth form phenotyping

We phenotyped worms in two culture conditions. Rearing *P. pacificus* on solid NGM induces the Eu morph and facilitates identification of Eu-deficient (all-St) phenotypes. Conversely, growing worms in liquid S-medium (5.85 g/L NaCl, 1 g/L $K_2HPO_4$, 6 g/L $KH_2PO_4$, 5 mg/L cholesterol, 3 mM $CaCl_2$, 3 mM $MgSO_4$, 18.6 mg/L disodium EDTA, 6.9 mg/L $FeSO_4 \bullet 7H_2O$, 2 mg/L $MnCl_2 \bullet 4H_2O$, 2.9 mg/L $ZnSO_4 \bullet 7H_2O$, 0.25 mg/L $CuSO_4 \bullet 5H_2O$ and 10 mM Potassium citrate buffer at pH 6.0) represses the Eu morph and facilitates identification of Eu-constitutive (all-Eu) phenotypes [19,71]. As food source, S-medium contained *E. coli* OP50 in the amount corresponding to 100 mL of an overnight culture with $OD_{600}$ 0.5 per 10 mL of medium. We started phenotyping by isolating eggs from stock culture plates, which contained large numbers of gravid hermaphrodites and eggs deposited on the agar surface [71]. To isolate eggs, we washed worms and eggs from plates with water, and incubated them in a mixture of 0.5 M NaOH and household bleach at 1:5 final dilution for 10 min with regular vortexing to disintegrate vermiform stages. Remaining eggs were pelleted at 1,300 g for 30 sec, washed with 5 mL of water, pelleted again, resuspended in water and pipetted on agar plates or into S-medium. Agar plates were left at room temperature (20–25˚C) for 3–5 days and 25 mL Erlenmeyer flasks with liquid medium were shaken at 22˚C, 180 rpm for 4–6 days. Adult hermaphrodites were immobilized on 5% Noble Agar pads with 0.3% $NaN_3$ added as an anaesthetic, and examined using differential interference contrast (DIC) microscopy. Animals that had a large right ventrosublateral tooth, curved dorsal tooth, and the anterior tip of the promesostegostom posterior to the anterior tip of the gymnostom plate were classified as Eu morphs. Animals that did not exhibit these three characters simultaneously were classified as St morphs, although there was a distinction between the morphology of *nhr-1* mutants and of other all-St mutants (S1A Fig).

## Geometric morphometric analysis

We reused the published [63] landmark data for the wild-type strain RS2333 and the *daf-21 (tu519)* mutant. We complemented this data set with newly collected data for the *nhr-1 (tu1163)* mutant, whereby we imaged young adults mounted on microscope slides on 5% Noble agar pads containing 0.3% $NaN_3$ as an anaesthetic. Only individuals with their right body side facing upwards were imaged. We took stack images of the anterior tip of the head, and recorded X and Y coordinates of 20 landmarks identical to the ones used in the previous study [63] using FIJI [72]. Procrustes alignment and PCA were done in R (ver. 3.4.4) [73] using geomorph package [74].

## CRISPR/Cas9 mutagenesis

We followed the previously published protocol for *P. pacificus* [75] with subsequently introduced modifications [76]. All target-specific CRISPR RNAs (crRNAs) were designed to target 20 bp upstream of the protospacer adjacent motifs (PAMs). We purchased crRNAs and universal trans-activating CRISPR RNA (tracrRNA) from Integrated DNA Technologies (Alt-R product line). 10 μL of the 100 μM stock of crRNA was combined with 10 μL of the 100 μM stock of tracrRNA, denatured at 95˚C for 5 min, and allowed to cool down to room

temperature and anneal. The hybridization product was combined with Cas9 protein (purchased from New England Biolabs or Integrated DNA Technologies) and incubated at room temperature for 5 min. The mix was diluted with Tris-EDTA buffer to a final concentration of 18.1 μM for the RNA hybrid and 2.5 μM for Cas9. When site-directed mutations were introduced via homology-directed repair, a ssDNA oligo template designed on the same strand as the gRNA was included in the mix at a final concentration of 4 μM. The diluted mixture was injected in the gonad rachis of approximately one day old adult hermaphrodites.

Eggs laid by injected animals within a 12–16 h period post injection were recovered, and the F1 progeny were singled out upon reaching maturity. After F1 animals have laid eggs, they were placed in 10 μL of single worm lysis buffer (10 mM Tris-HCl at pH 8.3, 50 mM KCl, 2.5 mM $MgCl_2$, 0.45% NP-40, 0.45% Tween 20, 120 μg/ml Proteinase K), frozen and thawed once, and incubated in a thermocycler at 65˚C for 1 h, followed by heat deactivation of the proteinase at 95˚C for 10 min. The resulting lysate was used as a template in subsequent PCR steps. Where possible, molecular lesions at the crRNA target sites were detected by melting curve analysis on a LightCycler 480 Instrument II (Roche) of PCR amplicons obtained using Light-Cycler 480 High Resolution Melting Master (Roche). Presence of mutations in candidate amplicons was further verified by Sanger sequencing. Alternatively, PCR was done using Taq PCR Master Mix (Qiagen) and all the F1 were Sanger sequenced.

To detect large rearrangements, we conducted whole genome re-sequencing of lines for which no PCR amplicon containing the crRNA target site could be obtained. For most such lines, we extracted genomic DNA using GenElute Mammalian Genomic DNA Miniprep Kit (Merck), whereby we modified the tissue digestion step by raising the Proteinase K concentration to 2 mg/mL, and prepared next-generation sequencing (NGS) libraries using Nextera DNA Flex Library Prep Kit (Illumina). For the *nhr-40* null mutant line, we followed a recently introduced cost-effective alternative procedure [77] with several modifications. Single worms were placed in 10 μL water, and frozen and thawed 3 times in liquid nitrogen. Then, we added 10 μL 2x single worm lysis buffer (20 mM Tris-HCl at pH 8.3, 100 mM KCl, 5 mM $MgCl_2$, 0.9% NP-40, 0.9% Tween 20, 240 μg/ml Proteinase K) and incubated the tubes in a thermocycler at 65˚C for 1 h. After a clean-up using HighPrep beads (MagBio Genomics), DNA was eluted in 7 μL Tris buffer at pH 8.0. Then, 100 pg of DNA was diluted with water to the total volume of 9 μL, mixed with 2 μL 5X TAPS-DMF buffer (50 mM TAPS at pH 8.5, 25 mM $MgCl_2$, 50% DMF) and 1 μL Tn5 transposase from Nextera DNA Library Prep Kit (Illumina) diluted beforehand 1:25 in dialysis buffer (100 mM HEPES at pH 7.2, 0.2 M NaCl, 0.2 mM EDTA, 0.2% Triton X-100, 20% glycerol). The mixture was incubated for 14 min at 55˚C. Tagmented DNA was amplified using Q5 HotStart High-Fidelity DNA Polymerase (New England Biolabs) for 14 cycles, whereby adapters and indices were added as primer overhangs, and size-selected for 250–550 bp fragments using HighPrep beads (MagBio Genomics). NGS libraries prepared using both methods were sequenced in a paired-end run of a HiSeq 3000 machine (Illumina). Reads were mapped to the El Paco assembly of the *P. pacificus* genome [78] using Bowtie 2 (ver. 2.3.4.1) [79]. We visually inspected read coverage in the loci of interest using IGV [80] to identify the precise regions in which coverage was close to zero.

## EMS mutagenesis

To induce heritable mutations in *P. pacificus*, we incubated a mixture of J4 larvae and young adults in M9 buffer (3 g/L $KH_2PO_4$, 6 g/L $Na_2HPO_4$, 5 g/L NaCl, 1 mM $MgSO_4$) with 47 mM ethyl methanesulfonate (EMS) for 4 h [81]. Subsequently, the worms were allowed to recover on agar plates with bacteria (see above), and 40–120 actively moving J4 larvae were singled out. After the animals have laid approximately 20 eggs, they were killed, and F1 progeny were

allowed to develop and reach maturity. F1 animals (which contained heterozygous mutants) were then singled out, and F2 progeny (which contained a mixture of genotypes, including homozygous mutants) were allowed to develop until adulthood. In each F1 plate, we determined the mouth form in 5–10 F2 individuals using Discovery V20 stereomicroscope (Zeiss). If at least one individual appeared to have a mouth form different from that of the background strain, such an animal was transferred to a fresh plate and its progeny was screened again using DIC until we gained confidence that a homozygous line was isolated. In the screen for suppressors of *nhr-40*, we mutagenized *nhr-40(tu505)* worms, which are all-Eu, screened approximately 1,000 F1 plates, and isolated one no-Eu allele, *tu515*. In an attempt to identify further downstream target genes, we conducted two suppressor screens in the *nhr-1(tu1163)* mutant background and screened approximately 3,800 F1 plates in total, but found no Eu individuals.

## Mapping of *tu515*

We crossed the *tu515* mutant, produced in the background of the RS2333 strain (a derivative of the PS312 strain), to a highly-Eu wild type strain PS1843. The resulting males were crossed to a strain RS2089, which is a derivative of PS1843 containing a morphological marker mutation causing the Dumpy phenotype. The progeny were allowed to segregate and 100 no-Eu lines were established. Four individuals from each line were pooled and genomic DNA was extracted from the pool using the MasterPure Complete DNA and RNA Purification Kit (Epicentre). Additionally, genomic DNA was extracted from the *tu515* line. NGS libraries were prepared using Low Input Library Prep kit (Clontech) and sequenced on Illumina HiSeq3000. Raw Illumina reads of the *tu515* mutant and of a mapping panel were aligned to the El Paco assembly of the *P. pacificus* genome (strain PS312) [78] by the aln and sampe programs of the BWA software package (ver. 0.7.17-r1188) [82]. Initial mutations were called with the samtools (ver. 1.7) mpileup command [83]. The same program was used to measure PS312 allele frequencies in the mapping panel at variant positions with regard to whole genome sequencing data of the PS1843 strain [78]. S2B Fig shows that large regions between the positions 5 Mb and 16 Mb of the *P. pacificus* chromosome X exhibit high frequency of the PS312 alleles (the mutant background) in the mapping panel. In total, 28 non-synonymous/nonsense mutations (S1 Table) in annotated genes (El Paco gene annotations v1, Wormbase release WS268) were identified in the candidate interval by a previously described custom variant classification software [84].

## Transgenesis

To identify putative promoter regions, which included 5' untranslated regions (UTR) and may have included the beginning of coding sequences, we manually re-annotated the 5' ends of predicted genes of interest using RNA-seq data and the information about predicted signal peptides. The ATG codon preceding the signal peptide or the last ATG codon in the second exon was designated as the putative start codon. As a general rule, the promoter region included a sequence spanning from the 3' end of the closest upstream gene on the same strand to the start codon, but if the upstream gene was located further than 2 kb away, a 1.5–2 kb region upstream of the identified start codon was designated as the putative promoter. In the case of inverted tandem duplicates in the head-to-head orientation, the 5' end of the promoter region was approximately in the middle between the start codons of the two genes. For the reporter constructs, we used the previously published coding sequences of TurboRFP [85] and Venus [27] fused with the 3' UTR of the ribosomal gene *rpl-23* [85]. For the *nhr-1* rescue construct, we used the native coding sequence, in which we replaced native introns with synthetic

introns, fused with the native 3' UTR. As the latter fragment could not be amplified from genomic or complementary DNA in one piece, we purchased a corresponding gBlocks fragments (Integrated DNA Technologies). FASTA sequences of all promoter regions, coding sequences and 3' UTRs are provided in S1 Data.

Plasmids carrying reporter and rescue constructs, listed in S2 Table, were created by Gibson assembly using NEBuilder HiFi DNA Assembly Master Mix (New England Biolabs) or a homemade master mix [86]. Small modifications, such as deletions and insertions under 70 bp, were introduced using Q5 Site-Directed Mutagenesis kit (New England Biolabs). Injection mix for transformation was created by digesting the plasmid of interest, the marker plasmid carrying a tail-bound reporter *egl-20p*::*TurboRFP* (if applicable), and genomic DNA with FastDigest restriction enzymes (Thermo Fisher Scientific), whereby genomic DNA was cut with an enzyme(s) that had the same cutting site(s) as the enzyme(s) used to digest the plasmids. Digested DNA was purified using Wizard SV Gel and PCR Clean-Up system (Promega), and the components were mixed in the following ratios. Injection mixes with rescue constructs contained 1 ng/µL rescue construct, 10 ng/µL marker, and 50 ng/µL genomic DNA. Injection mixes with reporter constructs contained 10 ng/µL reporter construct, 10 ng/µL marker, and 60 ng/µL genomic DNA. The mix was injected in the gonad rachis of approximately 1 day old hermaphrodites, and their progeny was screened for fluorescent animals [85].

## Antibody staining

We followed a previously published protocol [87] with minor modifications. Animals were washed from mature plates with phosphate-buffered saline (PBS) (137 mM NaCl, 2.7 mM KCl, 10 mM $Na_2HPO_4$, 1.8 mM $KH_2PO_4$ at pH 7.4), passed over a 5–20 µm nylon filter, concentrated at the bottom of a 2 mL tube and chilled on ice. We then added chilled fixative (15 mM Na-PIPES at pH 7.4, 80 mM KCl, 20 mM NaCl, 10 mM $Na_2EGTA$, 5 mM Spermidine-HCl, 2% paraformaldehyde, 40% MeOH), froze the worms in liquid nitrogen and thawed them on ice for 1–2 h with occasional inversion. Subsequently, the animals were washed twice with Tris-Triton buffer (100 mM Tris-HCl at pH 7.4, 1 mM EDTA, 1% Triton X-100), incubated in Tris-Triton buffer with 1% β-mercaptoethanol in a thermomixer at 600 rpm for 2 h at 37°C, washed once in borate buffer (25 mM $H_3BO_3$, 12.5 mM NaOH), incubated in borate buffer with 10 mM dithiothreitol in a thermomixer at 600 rpm for 15 min at room temperature, washed once in borate buffer, incubated in borate buffer with ~0.3% $H_2O_2$ in a thermomixer at 600 rpm for 15 min at room temperature, and washed once more in borate buffer. Next, the worms were washed three times with antibody buffer B (0.1% bovine serum albumin, 0.5% Triton X-100, 0.05% $NaN_3$, 1 mM EDTA in PBS) on a rocking wheel, incubated with a dye-conjugated antibody (Thermo Fisher Scientific, cat. # 26183-D550 and cat. # 26183-D488) diluted 1:25 in antibody buffer A (1% bovine serum albumin, 0.5% Triton X-100, 0.05% $NaN_3$, 1 mM EDTA in PBS) on a rocking wheel in the dark for 3 h at room temperature or overnight at 4°C, washed three times with antibody buffer B and mounted on slides in a 1:1 mixture of PBS and Vectashield (Vector Laboratories) with 1 µg/mL DAPI added. Slides were imaged using a Leica SP8 confocal microscope.

## RNA-seq analysis

To obtain a sufficient number of eggs, we passed young adult hermaphrodites to new agar plates with 5–10 animals per plate. After their F1 progeny have laid eggs (5–6 days), they were bleached (see above), then resuspended in 400 µL water per starting plate, pipetted onto multiple fresh plates with 100 µL suspension per fresh plate and placed at 20°C. Animals were collected at 24 h (corresponding to J2 and J3 larvae), 48 h (J3 and J4 larvae) and 68 h (J4 instar

larvae and young adults) post-bleaching by adding some water to the plates, scraping off the bacterial lawns with worms in them using disposable cell spreaders and passing the resulting suspension through a 5 μm nylon filter, which efficiently separated worms from bacteria. Worms were washed from the filter into 1.5 mL tubes, pelleted in a table-top centrifuge at the maximum speed setting, after which the supernatant was removed and 1 mL TRIzol (Invitrogen) was added to the worm pellets. Tubes were flash-frozen in liquid nitrogen and stored at -80˚C for up to a month. To extract RNA, worms suspended in TRIzol were frozen and thawed three times in liquid nitrogen, debris were pelleted for 10–15 min at 14,000 rpm at 4˚C, and 200 μL of chloroform was added to the supernatant. After vigorous vortexing and incubation at room temperature (20–25˚C) for 5 min, tubes were rotated for 15 min at 14,000 rpm at 4˚C. The aqueous phase was combined with an equal volume of 100% ethanol, RNA was purified using RNA Clean & Concentrator Kit (Zymo Research) and its integrity was verified using RNA Nano chips on the Bioanalyzer 2100 instrument (Agilent).

To analyse the transcriptome at the time of mouth form determination, we combined 500 ng RNA isolated at 24 h with 500 ng RNA isolated at 48 h post-bleaching, and proceeded to make libraries using NEBNext Ultra II Directional RNA Library Prep Kit for Illumina (New England Biolabs). To analyse the transcriptome at the time of mouth form differentiation, we prepared libraries from 1 μg of RNA isolated at 68 h post-bleaching. For wild type strain PS312, four biological replicates were collected at different time points. For the mutants, two replicates of two independent alleles were collected at two different time points, and these were treated as four biological replicates. Specifically, we sequenced the following alleles: *nhr-1 (tu1163) loss-of-function*, *nhr-1(tu1164) loss-of-function*, *nhr-40(tu505) gain-of-function*, *nhr-40(iub6) gain-of-function*, *nhr-40(tu1418) loss-of-function*, *nhr-40(tu1423) null*.

Libraries were sequenced in two paired-end runs of a HiSeq 3000 machine, whereby we aimed at 10–20 mln reads per library. Raw sequences have been deposited in the European Nucleotide Archive with the study accession number PRJEB34615 (http://www.ebi.ac.uk/ena/data/view/PRJEB34615). The fourth biological replicate of wild-type PS312 and all replicates of the *nhr-40 loss-of-function/null* mutants were sequenced in a different run than the other samples. To ensure that batch effects were negligible, we additionally re-sequenced the first three replicates of wild-type PS312 in the same run and verified that coordinates in PCA conducted using complete transcriptomes were minimally altered when comparing the same samples sequenced in the two runs. Reads were mapped to the El Paco assembly of the *P. pacificus* genome [78] using STAR (ver. 020201) [88]. Differential expression analysis was carried out in R (ver. 3.4.4) [73] using Bioconductor (ver. 3.6) [89] and DESeq2 (ver. 1.18.1) [90], whereby we counted reads mapping to El Paco v1 gene predictions [78]. We applied an adjusted p-value cutoff of 0.05 and no fold change cutoff. Alignments and coverage were visualized in IGV [80].

To examine the transcript levels of *nhr-1* and *nhr-40*, we repeated differential expression analysis, whereby we counted reads mapping to Trinity-assembled transcripts generated from previously published RNA-seq data [26] because the El Paco v1 gene prediction for *nhr-1* was incorrect in that it was a fusion of multiple neighboring genes. To test the differences in FPKM (Fragments Per Kilobase of transcript per Million mapped reads) values for *nhr-1* and *nhr-40* in different mutants at each of the two time points, we performed t-test as implemented in the t.test function in R (ver. 3.4.4) [73] and applied false discovery rate (FDR) correction to the p-values obtained. Prior to conducting the t-test, we verified the assumptions for parametric statistics by performing Shapiro-Wilk test for normality (shapiro.test function) and Levene test for homoscedasticity (levene.test function of the car package [91]). Signal peptides were predicted using SignalP (ver. 4.1) [92]. To compare relative numbers of genes in different categories listed in Fig 3B, we used chi-squared test as implemented in the chisq.test function in R (ver. 3.4.4) [73].

## Phylogenetic reconstructions

To identify NHR, CAP, and chitinase genes in the *C. elegans* genome, we retrieved the current version (PRJNA13758) of predicted proteins and domains from the http://wormbase.org website and selected genes that contained "IPR001628", "CAP domain", and "IPR001223" as predicted InterPro domains, respectively. The list of Astacin genes was taken from an earlier study [93] and the corresponding gene predictions were manually retrieved from the http://wormbase.org website. To identify NHR, Astacin, CAP, and chitinase genes in the *P. pacificus* genome, we predicted domains in the El Paco v1 version of gene predictions [78] using HMMER (ver. 3.1b2) software in conjunction with the PFAM profile database [94] and selected genes that contained "PF00105", "Astacin", "CAP", and "PF00704" as predicted PFAM domains, respectively. Manual inspection of the retrieved NHR genes in *P. pacificus* revealed that many of the gene predictions represent fusions of multiple neighboring genes. Therefore, we used the information about the predicted domains, RNA-seq data generated in this study, and Illumina and PacBio RNA-seq datasets generated earlier [26,95,96] to manually reannotate the NHR gene predictions in *P. pacificus*. We submitted the improved annotations to http://wormbase.org as part of a larger set of manually curated gene annotations [97]. For the tree of CAP domains in *P. pacificus* and *P. fissidentatus*, we predicted domains in the Pinocchio versions of gene predictions for both genomes [60] and selected genes that contained "PF00188" as a predicted PFAM domain. In the case of NHR genes, complete sequences were aligned, while in the case of other gene families, functional domains extracted using HMMER (ver. 3.1b2) were aligned to facilitate the alignment of genes with divergent domain architecture. Alignments were done in MAFFT (ver. 7.310) [98] and maximum likelihood trees were built using RAxML (ver. 8.2.11) [99]. Protein-based trees were generated with the following parameters: -f a -m PROTGAMMAAUTO -N 100. In the case of CAP domains in *P. pacificus* and *P. fissidentatus*, we first generated a protein-based tree and identified a poorly resolved subtree containing the genes of interest. To increase the number of informative sites, we extracted corresponding nucleotide sequences, aligned them in MAFFT and built a tree in RAxML with the following parameters: -f a -m GTRCAT -N 100. Obtained phylogenetic trees were visualized using FigTree (ver. 1.4.2). All phylogenetic trees and corresponding alignments are provided in S2 Data.

## Supporting information

**S1 Fig. Additional images.** (A) The mouth of the wild-type eurystomatous (Eu) morph, wild-type stenostomatous (St) morph, *nhr-1* mutant, and *nhr-40* mutant in two focal planes. (B) Expression pattern of an *nhr-1* transcriptional reporter in a young larva. TurboRFP channel is presented as a maximum intensity projection. (C) Antibody staining against the HA epitope in a line, in which the tag was "knocked in" into the endogenous locus. Fluorescent channel is presented as a maximum intensity projection. (D) Expression patterns of *nhr-40* and *nhr-1* transcriptional reporters in a double reporter line. TurboRFP (magenta) and Venus (green) channels are presented as standard deviation and maximum intensity projections, respectively. Co-expression results in white color. (E) Expression patterns of *nhr-40* and *nhr-1* transcriptional reporters in a double reporter line. TurboRFP is encoded as magenta, Venus as green. Co-expression results in white color. D = dorsal, V = ventral, A = anterior, P = posterior. (TIF)

**S2 Fig. Additional bioinformatic analyses.** (A) Whole-genome re-sequencing and RNA-seq of the *null* allele of *nhr-40*. (B) bulked segregant analysis of the suppressor of *nhr-40(tu505)* with the location of *nhr-1* marked with a dotted line. See S1 Table for the list of non-

synonymous and nonsense substitutions within the candidate region.
(TIF)

**S3 Fig. Geometric morphometric analysis of selected strains.** Geometric morphometric analysis of 20 landmarks in the mouth of the wild-type strain RS2333, the *nhr-1(tu1163)* mutant isolated in this study, and the *daf-21*/Hsp90*(tu519)* mutant previously shown to exhibit an aberrant mouth morphology while maintaining the dimorphism. Each point in the PCA plot corresponds to a single animal. Deformation grids at the extremes of the two axes display differences to the mean shape of all individuals.
(TIF)

**S4 Fig. The effect of *nhr-1* mutations on mouth morphology is epistatic to that of *nhr-40* mutations.** The mouth of the *nhr-1 loss-of-function* mutant, *nhr-40 null* mutant and the double *nhr-40 loss-of-function nhr-1 loss-of-function* mutant in two focal planes. D = dorsal, V = ventral, A = anterior, P = posterior.
(TIF)

**S1 Table. Candidate substitutions.** List of non-synonymous and nonsense substitutions within the candidate region on chromosome X identified through the bulked segregant analysis of the suppressor of *nhr-40(tu505)*.
(XLSX)

**S2 Table. Description of transgenic constructs.** Promoters include 5' UTRs and may include coding exons, and introns. See S1 Data for the sequences of the listed elements. CDS = coding sequence, UTR = untranslated sequence.
(XLSX)

**S1 Data. FASTA file of nucleotide sequences used to create transgenic constructs.**
(FAS)

**S2 Data. Phylogenetic trees from Fig 5 and alignments used to generate them.**
(ZIP)

**S3 Data. Data used to build the graphs in Fig 2B.**
(CSV)

**S4 Data. Data used to build the graphs in Fig 3B.**
(TXT)

**S5 Data. Landmark coordinates used for the analysis shown in S3 Fig.**
(TXT)

## Acknowledgments

We are grateful to Gabi Eberhardt and Tobias Loschko for their assistance with the mutagenesis screens and Jürgen Berger for taking the SEM image of the *P. pacificus* mouth. We thank Dr. Michael Werner, Dr. Neel Prabh, Dr. Adrian Streit, and Metta Riebesell for the discussion.

## Author Contributions

**Conceptualization:** Ralf J. Sommer.

**Data curation:** Bogdan Sieriebriennikov, Christian Rödelsperger, Ralf J. Sommer.

**Formal analysis:** Bogdan Sieriebriennikov, Christian Rödelsperger.

**Funding acquisition:** Ralf J. Sommer.

**Investigation:** Bogdan Sieriebriennikov, Shuai Sun, James W. Lightfoot, Hanh Witte, Eduardo Moreno.

**Methodology:** Bogdan Sieriebriennikov, Shuai Sun, James W. Lightfoot, Hanh Witte, Christian Rödelsperger.

**Project administration:** Ralf J. Sommer.

**Resources:** Hanh Witte.

**Software:** Bogdan Sieriebriennikov, Christian Rödelsperger.

**Supervision:** Ralf J. Sommer.

**Validation:** Shuai Sun, James W. Lightfoot, Eduardo Moreno.

**Visualization:** Bogdan Sieriebriennikov.

**Writing – original draft:** Bogdan Sieriebriennikov, Ralf J. Sommer.

**Writing – review & editing:** Bogdan Sieriebriennikov, James W. Lightfoot, Ralf J. Sommer.

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
