## [Decision Letter · Decision Letter 0]

8 Feb 2020

Dear Dr Sommer,

Thank you very much for submitting your Research Article entitled 'Conserved hormone-receptors controlling a novel plastic trait target fast-evolving genes expressed in a single cell' to PLOS Genetics. Your manuscript was fully evaluated at the editorial level and by independent peer reviewers. The reviewers appreciated the attention to an important topic but had a few, mostly minor suggestions for improving the manuscript.

We therefore ask you to modify the manuscript according to the review recommendations. Your revisions should address the specific points made by each reviewer.  Please note that although one of the reviewers suggested additional experiments that could reinforce some conclusions, it is up to you whether to add any more data to this already comprehensive and well-executed story.

[LINK]

Yours sincerely,

Artyom Kopp

Associate Editor

PLOS Genetics

Kirsten Bomblies

Section Editor: Evolution

PLOS Genetics

Reviewer's Responses to Questions

**Comments to the Authors:**

Reviewer #1: This is a beautiful study describing the genetic role of two nuclear hormone receptors in tooth formation in Prichtionchus pacificus. The quality of the data is exemplary, if only all the papers I review were of this quality!

I have only two small suggestions:

Page 9: “the lack of linear transcriptional regulation is consistent with different phenotypic effects of nhr-1 and nhr-40” – this is only a reasonable statement if one NHR would transcriptionally activate the other, not when it would repress. Remove or rephrase.

Page 13: the discussion on X-chromosome location of NHR target genes may be more appropriate in the Discussion of the paper.

Reviewer #2: See attached comments

Reviewer #3: In this manuscript by Sieriebriennikov et al., the authors further examine the mechanisms underlying the development and evolution a form of phenotypic plasticity regarding two mouth morphs of the nematode Pristionchus pacificus. To do this, the authors used a staggering combination of techniques and approaches, including mutagenesis, suppressor screens, genome editing, transgenesis, comparative transcriptomics, molecular phylogeny…

This is an impressive tour de force to address an important fundamental question of how plastic phenotypes originate and evolve.

The authors identified a nuclear receptor (nhr-1) that together with another NR (nrh-40) play a pivotal role in regulating mouth morphs (eurystomatous vs. stenostomatous). They went on to determine how these two genes interact and found that they share a number of downstream targets, consisting mostly of extracellular proteins (Astacin, CAP, etc..) that restricted to the Pristionchus taxon. They validated many of these targets using RNAseq data, and found that they were mostly expressed (together with the two nhr) during mouth form differentiation. They tried to determine the function of a sample of these targets in mouth morph determination, but no phenotypes were obtained through Crispr/Cas9 editing. Finally, the authors looked at the evolutionary history of the targets of nhr-1/40 and found that they were fast evolving and undergo rapid turn-over throughout worm phylogeny.

This work is substantial, rigorous and exciting. The sheer amount of data in this paper, that fits well into one complete story, makes it very rich and would bring an important addition to the field of phenotypic plasticity and gene network evolution. Furthermore, despite the large amount of information, the paper is so well written which makes it easy to follow.

The only minor thing I have is that I find the section in page 14 lines 271-283 a bit speculative about what the targets might be doing based on their GO terms.

The authors also tried hard to obtain phenotypes from the Astacin, chitinase, and CAP mutants without any luck. The authors explain this negative result by redundancy, which seems quite reasonable to me. I wonder if the authors observed any increase in expression of other members of Astacin, CAP, and chitinase in these mutants? This would confirm redundancy being the source for the lack of phenotypes.

Overall, this paper is exciting and well executed and written. As far as I am concerned, it is already in shape for publication.

**Have all data underlying the figures and results presented in the manuscript been provided?**

Reviewer #1: Yes

Reviewer #2: Yes

Reviewer #3: Yes

PLOS authors have the option to publish the peer review history of their article (what does this mean?). If published, this will include your full peer review and any attached files.

Reviewer #1: No

Reviewer #2: No

Reviewer #3: No

---

## [Editor Report · Decision Letter 1]

20 Feb 2020

Dear Dr Sommer,

We are pleased to inform you that your manuscript entitled "Conserved hormone-receptors controlling a novel plastic trait target fast-evolving genes expressed in a single cell" has been editorially accepted for publication in PLOS Genetics. Congratulations!

Yours sincerely,

Artyom Kopp

Associate Editor

PLOS Genetics

Kirsten Bomblies

Section Editor: Evolution

PLOS Genetics

Comments from the reviewers (if applicable):

**Data Deposition**

http://datadryad.org/submit?journalID=pgenetics&manu=PGENETICS-D-19-02036R1

**Press Queries**

---

## [Editor Report · Acceptance letter]

7 Apr 2020

PGENETICS-D-19-02036R1 

Conserved nuclear hormone receptors controlling a novel plastic trait target fast-evolving genes expressed in a single cell 

Dear Dr Sommer, 

We are pleased to inform you that your manuscript entitled "Conserved nuclear hormone receptors controlling a novel plastic trait target fast-evolving genes expressed in a single cell" has been formally accepted for publication in PLOS Genetics! Your manuscript is now with our production department and you will be notified of the publication date in due course.

With kind regards,

Jason Norris

PLOS Genetics

On behalf of:
